# Dense Residual Transformer for Image Denoising

**Chao Yao** [1,†] **, Shuo Jin** [2,†] **, Meiqin Liu** [2,*] **and Xiaojuan Ban** [3,*]

1. School of Computer and Communication Engineering, University of Science and Technology Beijing, Beijing 100083, China; yaochao@ustb.edu.cn
2. Institute of Information Science, Beijing Jiaotong University, Beijing 100044, China; 21125180@bjtu.edu.cn
3. Institute of Artificial Intelligence, University of Science and Technology Beijing, Beijing 100083, China
* Correspondence: mqliu@bjtu.edu.cn (M.L.); banxj@ustb.edu.cn (X.B.)
† These authors contributed equally to this work.

**Abstract:** Image denoising is an important low-level computer vision task, which aims to reconstruct a noise-free and high-quality image from a noisy image. With the development of deep learning, convolutional neural network (CNN) has been gradually applied and achieved great success in image denoising, image compression, image enhancement, etc. Recently, Transformer has been a hot technique, which is widely used to tackle computer vision tasks. However, few Transformer-based methods have been proposed for low-level vision tasks. In this paper, we proposed an image denoising network structure based on Transformer, which is named DenSformer. DenSformer consists of three modules, including a preprocessing module, a local-global feature extraction module, and a reconstruction module. Specifically, the local-global feature extraction module consists of several Sformer groups, each of which has several ETransformer layers and a convolution layer, together with a residual connection. These Sformer groups are densely skip-connected to fuse the feature of different layers, and they jointly capture the local and global information from the given noisy images. We conduct our model on comprehensive experiments. In synthetic noise removal, DenSformer outperforms other state-of-the-art methods by up to 0.06–0.28 dB in gray-scale images and 0.57–1.19 dB in color images. In real noise removal, DenSformer can achieve comparable performance, while the number of parameters can be reduced by up to 40%. Experimental results prove that our DenSformer achieves improvement compared to some state-of-the-art methods, both for the synthetic noise data and real noise data, in the objective and subjective evaluations.

**Keywords:** image denoising; residual skip connection; transformer

## 1. Introduction

The acquisition of images and videos is basically dependent on some digital devices; however, the collection process is often affected by various degradation factors. Lots of noise is blended into the images/videos, resulting in the loss of image quality. Therefore, image denoising aims to recover the clean image with high quality. Obviously, this problem is ill-posed, because the degradation process is unknown and irreversible. To tackle this problem, some early denoising methods utilize some specific filters, such as mean filters, nonlocal means (NLM) [1], and 3D transform-domain filtering (BM3D) [2], to eliminate the possible noise in the given noisy images. In recent years, some convolutional neural networks (CNN) have been also employed for image denoising. To explore how to further improve the denoising performance is still a hot challenge, especially for real scenarios.

With the development of deep learning, many state-of-the-art CNN models have achieved good performance in many computer vision tasks. Due to several revolutionary works, CNN-based networks have become the main benchmark method for image denoising. Some classical architectures such as residual skip connection [3] and dense skip connection [4] are utilized to improve the feature representation ability for image denoising.

Nevertheless, CNN-based networks showed that the fitting ability of the noise distribution could not be improved following with the increasing of CNN layers. Some technical issues such as gradient vanishing also plagued researchers. Moreover, for real image denoising, the performance of removing complex noise is also limited by only utilizing a residual learning strategy.

To tackle the above problems, on one hand, many works have attempted to employ the attention mechanism to enhance the representation of the local features, including channel attention and spatial attention. On the other hand, the global information is also considered to be an important addition to recover the clean images. So, the non-local operation is widely utilized in some of the latest network architectures. Specifically, Transformer [5] has been proved to be a useful tool for capturing the global information by utilizing the long-range dependencies of pixels, which is originally used in the natural language processing task. Alexey et al. propose the ViT [6] network, which successfully applies Transformer in computer vision tasks. Then, various exciting works [7–9] have made a lot of effort to design Transformer-based architectures for the specific tasks. However, some technique issues still exist and limit the application of Transformer. For example, border pixels of images can be hardly utilized in Transformer, because the adjacent pixels are out of the range. Transformer cannot capture local information well enough. Therefore, how to apply Transformer, especially for the low-level computer vision tasks, is still a challenge.

In this paper, we propose an end-to-end Transformer-based model for image denoising, which is named Dense residual Transformer (DenSformer). DenSformer is composed of a preprocessing module, a local–global feature extraction module, and a reconstruction module. Specifically, the preprocessing module is exploited to extract shallow features from input images. The local–global feature extraction module consists of several Sformer groups, and each Sformer group includes several ETransformer layers and one convolutional layer. Additionally, the residual skip connection is utilized to assemble these layers. Finally, the reconstruction module is utilized to restore the clean image. We quantitatively compare our DenSformer with other existing denoising methods. Experimental results on different test sets verify the effectiveness of our model in objective and subjective evaluation.

Overall, our contributions of this paper can be summarized as follows:

- We propose an end-to-end Transformer-based network for image denoising, where both Transformer and convolutional layers are utilized to implement the fusion between the local and global features.
- We design a residual in residual architecture to assemble multiple Transformers and convolutional layers to achieve better performance in a deeper network.
- We introduce a depth-wise convolutional layer into Transformer, which is used to preserve the local information in the forward process of Transformer.

## 2. Related Work

### 2.1. CNN-Based Image Denoising

With the development of deep learning, many researchers attempt to design novel denoising models based on convolution neural network (CNN), and most of them have achieved impressive improvement on the performance. Early CNN-based models are trained in the synthetic image data with AWGN (Additive White Gaussian Noise). Harold et al. propose to apply the original feedforward neural network model for the denoising task, which can achieve comparable results with BM3D [2]. Chen et al. [10] propose a trainable nonlinear reaction diffusion (TNRD) model to remove noise of different levels. Zhang et al. [11] design a named DnCNN Network, which demonstrates the availability of residual learning for image denoising, and Lin et al. [12] propose an Adaptive and Overlapped Average Filter (AOAF) to get better distribution of noise. Later on, more works are focused on the design of CNN-based network architectures [13], including the improvement of receptive field and the balance of performance and model size. These models can learn the noise distribution well from the training data, but they are not well suitable for the real noise removal. Then, Zhang et al. [14] further generate a noise map by making a relation between

the noise level and the noisy image, and they demonstrate a spatial-invariant denoising algorithm for real-image denoising. MIRNet [15] presents a network architecture that is able to handle multiple image restoration tasks including image denoising, super-resolution, and enhancement, with many new building blocks that could extract and utilize different feature information at multi-scale. RDN [4] combines DenseNet blocks and global residual learning to implement the fusion of local and global features. Although these models attempt to increase the performance of real denoising, it is still necessary to balance the trade-off between model size and performance.

### 2.2. Vision Transformer

Recently, Transformer [5,16] has been popular in the computer vision field. The emergence of ViT [6,17] marks the beginning of applying Transformer in the computer vision field. Nevertheless, there are still some weaknesses, such as the requirement for large-scale datasets and the high time complexity. Driven by ViT, various Transformer-based models are proposed to solve the problems related to computer vision tasks, such as image classification [9], image/video segmentation [18,19], and object detection [8,20] etc. Some of the latest works have tried to apply Transformer for image restoration. Wang et al. propose a Detail-Preserving Transformer [21] for the light field image super-resolution, which can effectively restore the detailed information of light field images by using Transformer. Chen et al. [22] propose a backbone model named IPT, which is based on the standard Transformer, to restore multiple image degradation tasks. However, IPT is a pre-training model, which means it need to be firstly trained on a large-scale dataset, and the performance of image restoration is also limited. Wang et al. [23] further propose a U-shaped Transformer network, and it achieves good performance on image restoration. However, how to apply Transformer to solve the problem of the low-level computer vision tasks is still a hot issue.

## 3. Proposed Work

In this section, we first describe our designed network architecture. Then, some details of the main components in DenSformer are provided. Finally, we discuss the effects of the residual skip-connection strategies on the fusion between the local and global features.

### 3.1. Network Architecture

As shown in Figure 1, DenSformer is composed of the preprocessing module, the local-global feature extraction module, and the reconstruction module. The preprocessing module contains a $3 \times 3$ convolutional layer to extract shallow features from the input image $I_{in} \in \mathbb{R}^{H \times W \times C_{in}}$ (Naturally, $H$, $W$ are the height and weight of the image , and $C_{in}$ is the channel number of the given image). Then, the shallow features $F_0 \in R^{H \times W \times C}$ can be obtained, where $C$ is the channel number of the shallow feature. It can be represented as,

$$F_0 = H_{pre}(I_{in}), \tag{1}$$

where $H_{pre}$ denotes the preprocessing module.

Next, a local–global feature extraction module is designed to extract the local and the global features from $F_0$, where the features are learned by multiple Transformers (named Sformer Block) in a residual-in-residual. These Transformers are assembled by some dense skip connections. In addition, long skip connections are exploited. So, we can further have

$$F_D = H_{TB}(F_0), \tag{2}$$

$H_{TB}(\cdot)$ denotes our proposed dense Transformer-based structure, which contains $M$ Sformer blocks. We treat the output $F_D$ as deep features, which is then converted to the high-quality image $I_{out}$ by concatenating with $F_0$ with a long skip connection

$$I_{out} = H_{rec}(F_D) + F_0 = H_{DenS}(I_{in}), \tag{3}$$

where $H_{res}$ denotes the reconstruction module, which is made up by a $3 \times 3$ convolutional layer. With dense residual skip connection, our network can transmit the shallow layer features directly to the reconstruction module. We use residual learning to reconstruct the residual between the noisy image and the corresponding clean image. $H_{DenS}$ is the mathematical representation of our DenSformer.

To improve the effectiveness of our DenSformer in the image denoising task, we choose $L_1$ loss to optimize the whole network. The training set $\{I_{noisy}, I_{clean}\}^N$ contains $N$ pairs of noisy images and the corresponding clean images. The training goal of DenSformer is to minimize the $L_1$ loss function [24],

$$L(\Theta) = \frac{1}{N} \sum_{i=1}^{N} \|H_{DenS}(I_{noisy}) - I_{clean}\|_1, \tag{4}$$

where $\Theta$ denotes the parameters set of our network. The loss function is optimized by using stochastic gradient descent. More details of our network can be found as below.

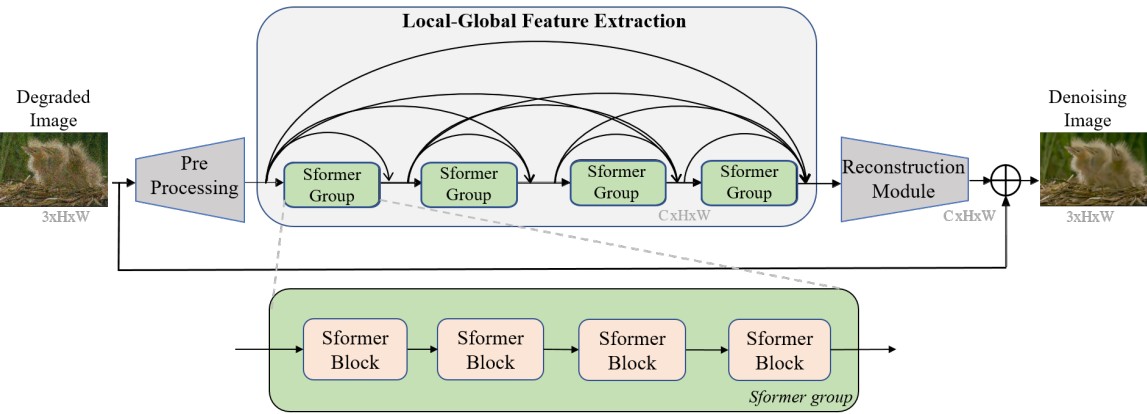

**Figure 1.** Architecture of the DenSformer.

### 3.2. Sfomer Block

We now give more details about our proposed Sformer block. As shown in Figure 2a, the Sformer block contains $M$ Enhanced Lewin Transformer (ETransformer) layers, a $3 \times 3$ convolutional layer behind all ETransformer layers, and a long skip connection bridging the input and output of the block. In Sformer blocks, each ETransformer layer will extract the deep feature from the input, and the behind convolution layer fuses the features, which will help the next deep feature extraction.

ETransformer is designed depending on LeWin Transformer [23]. To enhance the learning ability of local and global information in Transformer, the 2D input token is firstly reshaped to 3D along the spatial dimension and then processed by a depth-separable convolution. Instead of generating by direct linear layer operation, some local information can be preserved by the convolutional operation. Thereby, the **Q**, **K**, and **V** are changed as:

$$\mathbf{Q} = LN(H_{DS}(W_Q X_{i-1})) \quad \mathbf{K} = LN(H_{DS}(W_K X_{i-1})) \quad \mathbf{V} = LN(H_{DS}(W_V X_{i-1})), \tag{5}$$

where $H_{DS}(\cdot)$ denotes the depth-wise convolution layer, and $LN(\cdot)$ denotes the layer normalization. It is noted that the convolutional operation can also decrease the memory consuming in the process of calculating self-attention [25,26], where the spatial resolution of input features can be reduced. Otherwise, we perform the self-attention within non-overlapping local windows, which can further reduce the computational cost significantly.

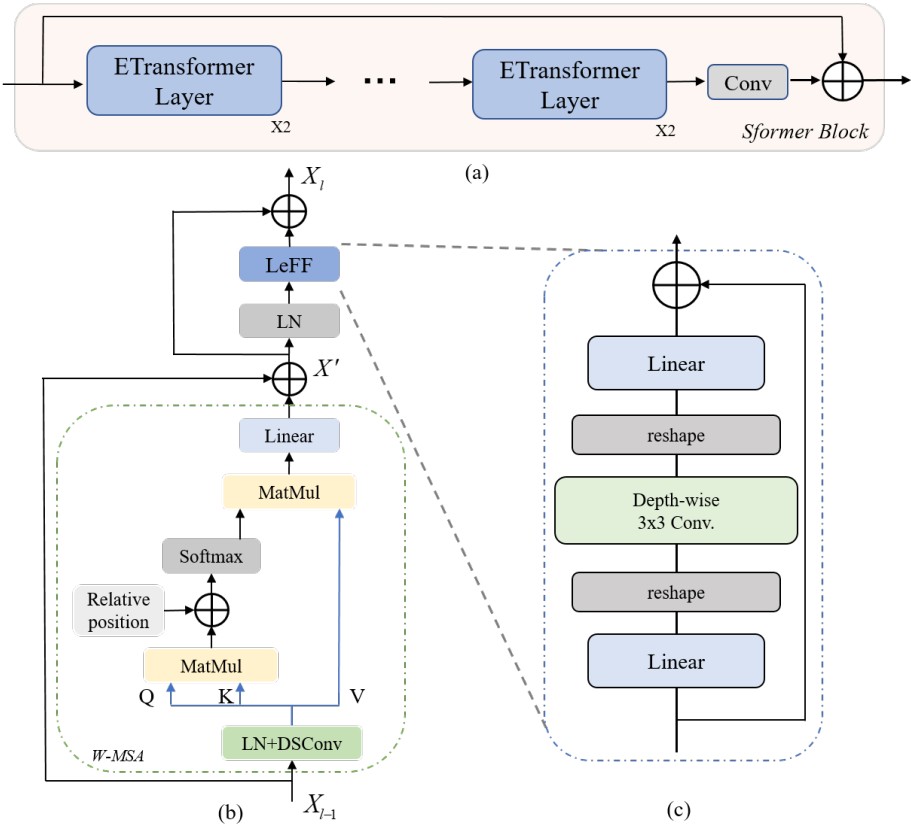

**Figure 2.** (**a**) Sformer block. (**b**) Architecture of the ETransformer. (**c**) Structure of the LeFF.

Given an input feature with size $W \times H \times C$, W-MSA first reshapes it to $HW \times C$ through the above convolutional layer, and the reshaped feature is further clipped by several $M \times M$ windows. The input feature is fed into the W-MSA with $\frac{HW}{M^2} \times C \times M^2$, where $\frac{HW}{M^2}$ is the total number of windows. Then, the self-attention weights are calculated separately window by window. Window-based self-attention can significantly reduce the computational cost compared with global self-attention. Given the feature map $X \in \mathbb{R}^{H \times W \times C}$, the computational complexity drops from $O(H^2 W^2 C)$ to $O(\frac{HW}{M^2} M^4 C) = O(M^2 HWC)$. We also include a relative position bias $B \in \mathbb{R}^{M^2 \times M^2}$ to each head in computing similarity, so the self-attention can be calculated as,

$$Attention(Q, K, V) = SoftMax(\frac{QK^T}{\sqrt{d}} + B)V, \qquad (6)$$

where $B$ is the learnable relative position encoding. Comparing with the vanilla self-attention module such as ViT [6], the computational complexity of the vanilla self-attention is $O(H^2 \times W^2 \times C)$ with the input $X \in \mathbb{R}^{(H \times W \times C)}$. By utilizing the window self-attention (W-MSA), since the self-attention weights are calculated in each small window, the computational complexity can drop to $O(M^2 HWC)$. The computational cost is highly optimized from quadratic complexity to linear complexity corresponding to the image size.

As shown in Figure 2c, a structure named LeFF will substitute the MLP layer in Feed-Forward Network (FFN) in order to avoid extracting local features insufficiently and compress memory. The Feed-Forward Network (FFN) in the standard Transformer presents limited capability to leverage local context. However, adjacent pixels are crucial references for image restoration. To further enhance the local information, we utilize LeFF [23] in substitution for MLP in the original Transformer. As shown in Figure 2c, a linear layer is firstly used to increase the feature dimension, so the output tokens can be reshaped to 2D feature maps. In the following, a $3 \times 3$ depth-wise convolution [27] is cascaded to capture

local information; then, we flatten the features maps to tokens again, and another linear layer is used to adjust the channels for matching with the input of the next ETransformer in the dimension. In this way, the depth-wise convolution in LeFF will help enhance the local information extraction with avoiding heavy computation.

Finally, the computation of our ETransformer layer is represented as:

$$
\begin{aligned}
X_l' &= W - MSA(lN(X_{l-1})) + X_{l-1} \\
X_l &= LeFF(lN(X_{l-1})) + X_l',
\end{aligned}
\tag{7}
$$

where $X_l'$ and $X_l$ are the outputs of the W-MSA module and *LeFF* module, respectively. $lN$ represents the layer normalization. On one hand, the convolutional operation in ETransformer layers can decrease the memory consuming in the process of calculating self-attention. On the other hand, local information will be extracted and then fused with the learned global information in ETransformer layers. Overall, the Sformer block can be formulated as follows:

$$
F_{D_i} = H_{TC}(H_{ET}(F_i)) + F_i = H_S(F_i) + F_i,
\tag{8}
$$

where $F_i$ denotes the input feature, $F_{Di}$ denotes the output of the $i$-th Sformer block, $H_{TC}(\cdot)$ denotes the convolution layer in the Sformer block, $H_{ET}(\cdot)$ denotes ETransformer layers, and $H_S(\cdot)$ denotes the whole Sformer block. With Sformer blocks, our network can extract deep features from the input and help reconstruct high-frequency features.

### 3.3. Dense Residual Skip Connection

With the emergency of ResNet [3] and DenseNet [28], skip connection in neural network has been proved to make the network more robust and stable training. Skip connection is also able to facilitate the fusion of feature information between different layers, especially when crossing multiple layers. Furthermore, applying dense residual-connection between multiple layers can make a better promotion on the performance. Generally speaking, shallow features mainly contain low-frequency information, while deep features focus on recovering the high-frequency information. With a number of long or short skip connections, the low and high-frequency information can be aggregated. In DenSformer, some convolutional operation and multi-head self-attention mechanism are exploited to capture the local and global features, respectively. Therefore, to better implement the fusion of the local–global information, we present a dense residual skip connection to better bridge the local and global features.

Combining schemes of skip connection variants, we design a skip-connection structure named dense residual skip connection. To fuse the shallow features and deep features, we add a skip connection between every two Sformer groups and a long skip connection between the shallow features to the deep features. All Sformer groups are densely skip-connected to enhance the feature representation. As shown in Figure 1, each Sformer group is connected to all the following groups. In this way, shallow features and deep features can be densely fused, where low and high-frequency information can be well reconstructed.

## 4. Experiments

In this section, we demonstrate the quantitive effectiveness of our method on both synthetic datasets and real noisy datasets; moreover, some visual results of denoised images are also provided to evaluate the subjective performance of our models.

### 4.1. Experimental Settings

**Training Data.** We use DIV2K and Flickr2K as our training dataset. The DIV2K dataset is a high-quality dataset that consists of 800 training images, 100 validation images, and 100 test images. The Flickr2K dataset contains 2650 images with high resolution. In the synthetic denoising experiment, different levels of AWGN [29] are added to the clean image for generating multiple degraded images. For real image denoising, we adopt the

*SIDD* medium dataset [27] for training. *SIDD* utilized five diverse mobile phones to take 30,000 noisy images in different scenes. For the clean image, *SIDD* removes the wrong pixels in each image, then aligns the image, and finally generates a "noiseless" real image.

**Testing Data and Metrics**. For synthetic noise removal, we adopt *Kodak24, CBSD68, and McMaster* as test sets for color image denoising and *Set12, BSD68, and Kodak24* as test sets for gray-scale image denoising. For real noise removal, we adopt the *SIDD* [27] validation dataset and *DnD* dataset [30]. In synthetic noise removal experiments, we compare DenSformer with other existing denoising methods including DnCNN [11], IRCNN [31], MemNet [32], FFDNet [14], RDN [4], etc. In real noise removal experiments, we compare DenSformer with BM3D [2], CBDNet [33], RIDNet [34], MPRNet [35], etc. Widely used quality assessments PSNR and SSIM are utilized to evaluate the performance of denoising. The best and second-best results are highlighted in **bold** and <u>underline</u> respectively.

**Implementation Details**. For different denoising tasks, the Sformer groups, Sformer blocks, and ETransformer layers are all set to 4. The number of feature channels is 64, and the patch size of the input image is $40 \times 40$. In the training stage, we employ ADAM [36] optimizer with the $\beta_1 = 0.9$, $\beta_2 = 0.999$ and $\epsilon = 10^{-8}$. The initial learning-rate is set to $1 \times 10^{-4}$ and the strategy of decreasing the learning rate is multistep decreasing, where the learning rate would decrease half for every 20K iterations. We apply rotation, cropping, and flipping on the training images to augment the training data. We conduct experiments on PyTorch and two NVIDIA TITAN XP GPUs.

*4.2. Synthetic Noisy Images*

To achieve a fair comparison, the same noise, AWGN, with different noise levels, $\sigma = 30, 50$, and 70, is added into the high-quality images to get noisy images. The final results of gray-scale and color images are listed in Tables 1 and 2, respectively. As shown in Table 1, for synthetic noise removal of gray-scale images, our method outperforms other image denoising methods and achieves the best performance in *Set12* and *Kodak24* test sets, except when the noise level $\sigma = 50$ in the *BSD68* test set. Take $\sigma = 30$ as an example; our DenSformer obtains 0.03 dB, 0.24 dB, 0.17 dB improvements over RDN [4] in all of three test set, respectively. In case of color image, as shown in Table 2, for instance when $\sigma = 30$, our network obtains 0.57 dB, 0.59 dB, and 1.19 dB improvements over RDN [4] in three test sets, respectively. The result proves the effectiveness of our proposed DenSformer on denoising the synthetic noise. It is noted that when the noise level is $\sigma = 70$, our network performs not well enough and worse than RDN. We analyze that it might be that the dense skip connection is not suitable for a high noise level situation. This is the point that we will fix in the future work.

**Table 1.** Quantitative results about non-blind gray-scale image denoising (PSNR (dB)).

| Method | Set12 | | | Kodak24 | | | BSD68 | | |
|---|---|---|---|---|---|---|---|---|---|
| | **30** | **50** | **70** | **30** | **50** | **70** | **30** | **50** | **70** |
| BM3D [2] | 29.13 | 26.72 | 25.22 | 29.13 | 26.99 | 25.73 | 27.76 | 25.62 | 24.44 |
| TNRD [10] | 28.63 | 26.81 | 24.12 | 28.87 | 27.20 | 24.95 | 27.66 | 25.97 | 23.83 |
| IRCNN [31] | 29.45 | 27.14 | N/A | 29.53 | 27.45 | N/A | 28.26 | 26.15 | N/A |
| DnCNN [11] | 29.53 | 27.18 | 25.52 | 29.62 | 27.51 | 26.08 | 28.36 | 26.23 | 24.90 |
| MemNet [32] | 29.63 | 27.38 | 25.90 | 29.72 | 27.68 | 26.42 | 28.43 | 26.35 | 25.09 |
| MWCNN [13] | N/A | <u>27.74</u> | N/A | N/A | N/A | N/A | N/A | **26.53** | N/A |
| FFDNet [14] | 29.61 | 27.32 | 25.81 | 29.70 | 27.63 | 26.34 | 28.39 | 26.30 | 25.04 |
| RDN [4] | <u>29.94</u> | 27.60 | <u>26.05</u> | <u>30.00</u> | <u>27.85</u> | <u>26.54</u> | <u>28.56</u> | 26.41 | <u>25.10</u> |
| DenSformer | **29.97** | **27.88** | **26.24** | **30.24** | **27.91** | **26.76** | **28.73** | <u>26.52</u> | **25.24** |

**Table 2.** Quantitative results about non-blind color image denoising (PSNR (dB)).

| Method | Kodak24 | | | BSD68 | | | McMaster | | |
|---|---|---|---|---|---|---|---|---|---|
| | **30** | **50** | **70** | **30** | **50** | **70** | **30** | **50** | **70** |
| BM3D [2] | 30.89 | 28.63 | 27.27 | 29.73 | 27.38 | 26.00 | 30.71 | 28.51 | 26.99 |
| TNRD [10] | 28.83 | 27.17 | 24.94 | 27.63 | 25.96 | 23.83 | N/A | 27.35 | N/A |
| IRCNN [31] | 31.24 | 28.93 | N/A | 30.22 | 27.86 | N/A | 31.11 | 28.91 | N/A |
| DnCNN [11] | 31.39 | 29.16 | 27.64 | 30.40 | 28.01 | 26.56 | 30.82 | 28.62 | 27.10 |
| MemNet [32] | 29.67 | 27.65 | 26.40 | 28.39 | 26.33 | 25.08 | N/A | 27.63 | 26.11 |
| FFDNet [14] | 31.39 | 29.10 | 27.68 | 30.31 | 27.96 | 26.53 | N/A | 29.18 | 27.66 |
| RDN [4] | 31.94 | 29.66 | 28.20 | 30.67 | 28.31 | **26.85** | 32.39 | 30.19 | **27.98** |
| DenSformer | **32.51** | **30.07** | **28.59** | **31.56** | **28.94** | 26.46 | **33.58** | **31.16** | 27.49 |

We also show visual denosing results of different methods. Specifically, the building textures and the chicken feathers are presented in Figures 3 and 4 respectively, which are hard to be separated with heavy noise. In the subjective results, we can find that other denoising methods tend to remove the edge details and the noise together, which make the results too smooth. BM3D [2] preserves the image structure to some degree but fails to remove noise deeply, DnCNN [11] and IRCNN [31] would make the edges of images over-smoothed, and FFDNet [14] is slightly better than the former two methods. RDN [4] restores more clean images, but there are still some textures and tiny details not restored well during the denoising process. The main reason is that these methods are limited to extract high-frequency features. On the contrary, our network is able to reconstruct local detail high-frequency features and clean smooth areas, because Enhanced Transformer in our method can better capture global information of the image, and the dense residual skip-connection scheme can fuse the shallow and deep features to help reconstruction.

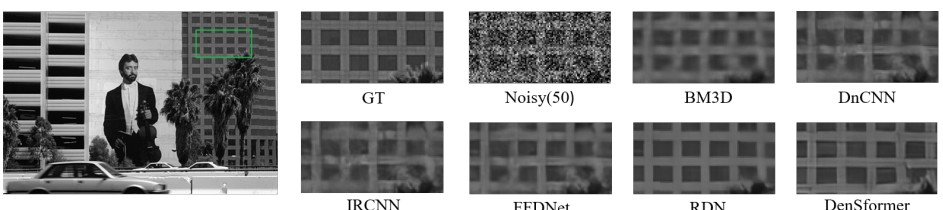

**Figure 3.** Gray-scale image denoising results with noise level $\sigma$ = 50.

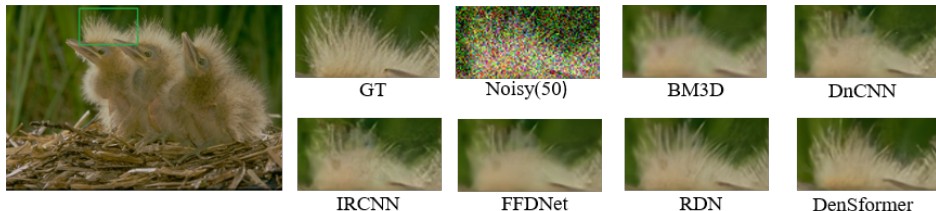

**Figure 4.** Color image denoising results with noise level $\sigma$ = 50.

### 4.3. Real Noisy Images

To evaluate the denoising performance of our method on real noisy images, we conduct a series of experiments. We adopt *SIDD* datasets as our train dataset, *SIDD* as our validation dataset, and the *DnD* dataset as our test dataset, which contains 1280 noisy-clean image pairs whose resolutions are $256 \times 256$.

In comparative experiments, the quantitative results on the test set and parameter comparison are shown in Table 3. Compared with VDN [37], which has the same parameters, DenSformer can achieve excellent PSNR performance that surpasses 0.40 dB on the *SIDD* dataset and 0.49 dB on the *DnD* dataset. That means our network has the ability to extract and reconstruct detailed information. Structural similarity index measure (SSIM), an image

quality assessment based on the degradation of structural information, is also employed to evaluate the performance of denoising. It can be observed that the metric SSIM of our model is almost consistent with recent state-of-the-art methods, which denotes that our DenSformer is effective at maintaining the consistency of the image's structure with fewer parameters. Although the PSNR result of MPRNet and Uformer is better than DenSformer, their parameters are 20.1 M and 20.6 M, along with our 7.8 M. Our light models can be better applied in real devices. Meanwhile, our network does not perform better on real image denoising, and we also analyze that it might be that DenSformer is not suitable for the images with huge size and unknown degradation. We will fix this challenge in future work.

**Table 3.** Quantitative results about real blind image denoising.

| Method | Parameters (MB) | SIDD Dataset | | Dnd Dataset | |
|---|---|---|---|---|---|
| | | PSNR (dB) | SSIM | PSNR (dB) | SSIM |
| BM3D [2] | - | 25.65 | 0.685 | 34.51 | 0.851 |
| CBDNet [33] | 4.3 | 30.78 | 0.801 | 38.06 | 0.942 |
| RIDNet [34] | 1.5 | 38.71 | 0.951 | 39.26 | 0.953 |
| VDN [37] | 7.8 | 39.28 | 0.956 | 39.38 | 0.952 |
| SADNet [38] | 4.3 | 39.46 | 0.957 | 39.59 | 0.952 |
| MPRNet [35] | 20.1 | <u>39.71</u> | 0.958 | 39.80 | 0.954 |
| Uformer [23] | 20.6 | **39.77** | **0.970** | **39.96** | **0.956** |
| DenSformer | 7.8 | 39.68 | <u>0.958</u> | <u>39.87</u> | <u>0.955</u> |

Figures 5 and 6 show the subjective comparison results of each method on the *SIDD* and *DnD* dataset, respectively. From the visualization results, we can find that the traditional algorithm BM3D is not good at denoising the real noise. While CBDNet can remove part of the real noise, the image after denoising is very blurred, and the image restored by RIDNet is too smooth, and the details are not clear enough. The image restored by our method is close to the results of other excellent algorithms such as MPRNet and Uformer, which can restore the image more clearly. As shown in Figure 7, DenSformer achieves competitive quantitive results with fewer parameters, which has a good balance between the PSNR result and model size.

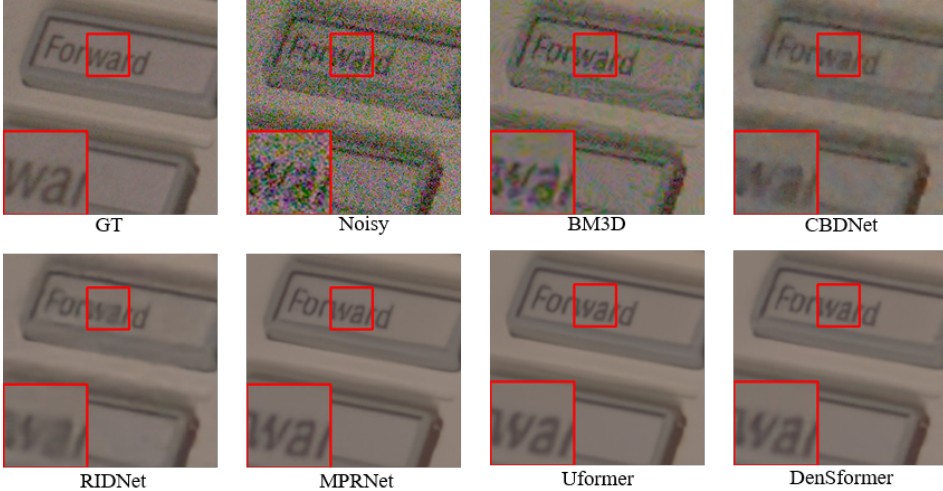

**Figure 5.** Denoising results of different existing methods from *SIDD*.

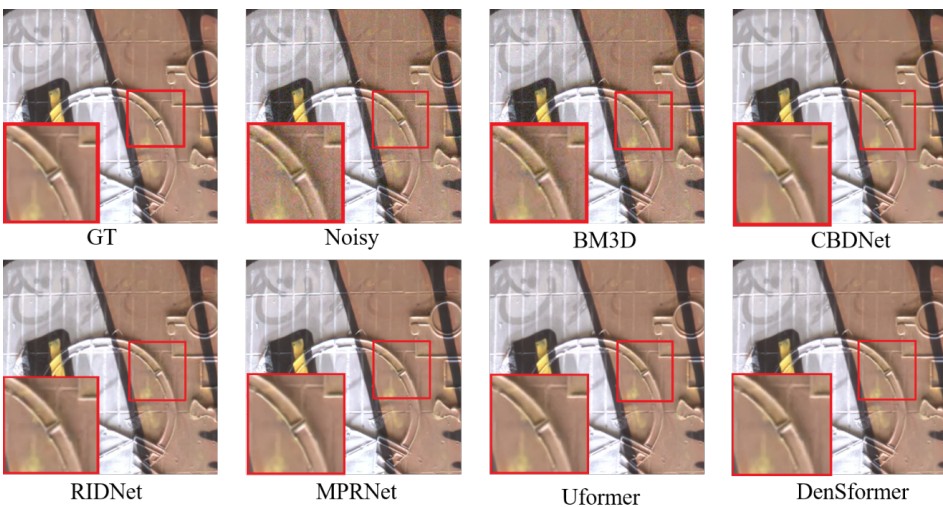

**Figure 6.** Denoising results of different existing methods from *DnD*.

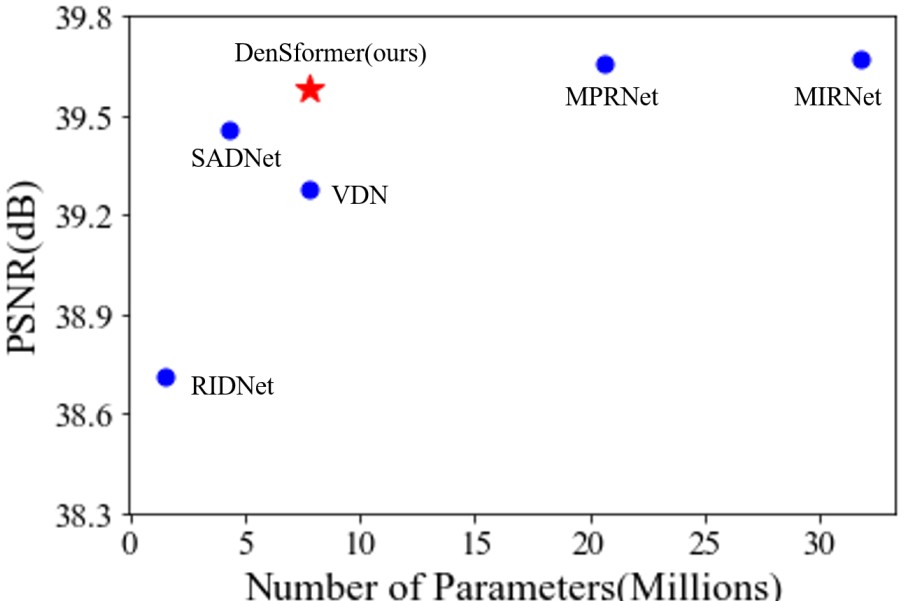

**Figure 7.** PSNR results v.s. number of parameters of different methods for denoising on SIDD.

### 4.4. Ablation Study

For ablation study, we train our DenSformer on *DIV2K* and *Flickr2K* and test it on *Kodak24*. We firstly compared three variants of skip connection in our network, including dense residual skip connection, local residual skip connection, and cross-residual skip connection. Specifically, dense residual skip connection is what we employed in DenSformer. Local residual skip connection means there are only residual skip connections between every two Sformer groups and a long skip connection between the input and output. Cross-residual skip connection is the scheme of dense residual skip connection but without local residual skip connection.

We show the effects of different schemes of skip connection in Table 4. It can be observed that the result of dense residual skip connection is the best, which outperforms 0.04 and 0.05 dB in PSNR compared with the other two schemes, respectively. Meanwhile, there is also a slight increase in training time, which means that our dense residaul skip-connection scheme will get a high PSNR performance only with a small time cost.

**Table 4.** Ablation study of FFN layer on *Kodak24*.

| Model | Local Connection | Global Connection | Cross Connection | PSNR | Training Time /Epoch |
|-------|------------------|-------------------|------------------|------|----------------------|
| Cross | ✓ | - | ✓ | 30.03 | 1.84 h |
| Local | ✓ | ✓ | - | 30.02 | 1.95 h |
| Dense | ✓ | ✓ | ✓ | 30.07 | 2.00 h |

Effects of different layers and blocks in ETransformer are also shown in Table 5. The implementation of experimental details is the same as in the ablation study of skip connection. The Vanilla model denotes the original Transformer layer with Window-based Multi head self-attention, the Vanilla-C model denotes the original Transformer in which Depth-wise convolution substitutes a linear layer in the generation of **Q**, **K**, **V**, LeWin means LeWin Transformer [23], and Enhanced LeWin is the original LeWin Transformer with depth-wise convolution. We can observe that the results of the Vanilla-C model are a little lower than those of the Vanilla model, and the results of the Enhanced LeWin model are better than those of the other three models, which means that the convolution layer in the generation of **Q**, **K**, and **V**, and the LeFF layer in the Feed-Forward Network can help Transformer better extract and utilize information, which can perform well in different denoising tasks.

**Table 5.** Ablation study of FFN layer on *Kodak24*.

| Model | DSConv | Linear | MLP Layer | LeFF Layer | PSNR |
|-------|--------|--------|-----------|------------|------|
| Vanilla | - | ✓ | ✓ | - | 29.86 |
| Vanilla-C | ✓ | - | ✓ | - | 29.84 |
| LeWin | - | ✓ | - | ✓ | 29.95 |
| Enhanced LeWin | ✓ | - | - | ✓ | 30.07 |

To further evaluate the impact of ETransformer layers in Sformer, we conduct another ablation experiment about the number of ETransformer layers. The results are summarized in Figure 8. It can be observed that the PSNR result is positively correlated with the number of ETransformer layers in the Sformer block. Nevertheless, the performance gain becomes saturated gradually with layers increasing. Meanwhile, the number of parameters will also increase and make our model huge. To balance the trade-off between parameters and performance, we choose four as the number of ETransformer layers.

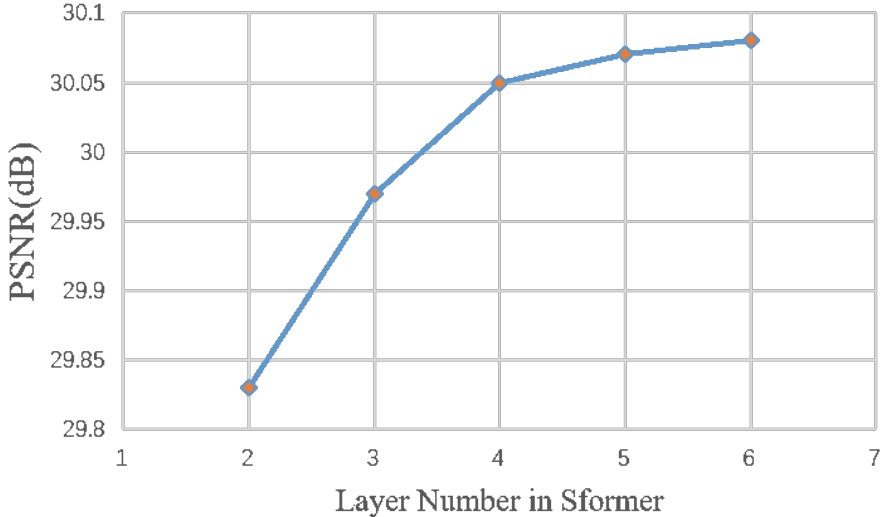

**Figure 8.** Impact of ETransformer layers in Sformer. Results are tested on *Kodak* with $\sigma = 50$.

## 5. Conclusions

In this paper, we proposed a new dense residual skip-connection network based on Transformer (DenSformer) for effective image denoising. Our DenSformer is made up of a preprocessing module, global feature extraction module, and reconstruction module. To better extract the local and global features and reconstruct the details, we design the Enhanced LeWin Transformer and take advantage of dense residual skip connection. Meanwhile, we employ dense residual skip connection among the Sformer groups to extract the deep features and bridge the information from different layers. The results on synthetic images with AWGN demonstrated that our DenSformer can achieve better results not only in gray-scale images but also color images compared with other existing methods. In addition, we have a good balance between experimental results and model size in real noise removal experiments. We believe that dense residual skip connection and Enhanced Transformer can help better fuse local–global features and achieve better results in image denoising and other restoration tasks.

**Author Contributions:** Conceptualization, C.Y. and S.J.; methodology, C.Y. and S.J.; supervision, M.L.; project administration, X.B.; funding acquisition, C.Y., M.L. and X.B. All authors have read and agreed to the published version of the manuscript.

**Funding:** This research was funded by the Fundamental Research Funds for the Central Universities (2019JBM018, FRF-TP-19-015A1, FRF-IDRY-20-038) and the National Natural Science Foundation of China (61972028, 61902022).

**Conflicts of Interest:** The authors declare no conflict of interest.

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
