# Peer review of "Dense Residual Transformer for Image Denoising"

_electronics, doi:10.3390/electronics11030418_

Round 1

Reviewer 1 Report

The authors have proposed an end-to-end Transformer-based network for image denoising. As mentioned in the paper, Transformers are slowly replacing CNNs in computer vision offering improved efficiency. The results show improved performance when compared to existing methods. For these reasons, this paper will be of interest to readers of this journal.

The paper is organized and well written with necessary figures and tables. Yet, following minor corrections/changes need to be considered.

  1. The quality metric SSIM is not explained anywhere in the manuscript. Please explain.
  2. What is the reason for adding synthetic images with AWGN rather than other noise types?
  3. Line 295:"...DenSformer is not suitable for images with huge size and unknown degradation". How can DenSformer improve its performance if the degradation is known?

Reviewer 2 Report

I reviewed the paper throughly and my opinion is that paper can become publishable after minor revision requirements that I listed below:

  1. Abstract should provide a summary of quantitative results based on comparisons with state of the art methods.
  2. I do not think there is need for Eq1 to define noisy image. Just writing your definition will be enough
  3. Figure 1 location is not appropriate and also it is not referenced in the text.
  4. Line 73, there is a need for revision of the sentence for clarity.
  5. Figure 2b is not a complex block structure so it can be combined with 2a.
  6. Line 144, why L1 loss function? Needs reference of better performance or quantification of use.
  7. Figure 4 is providing a replicated info that already presented in FÄ°gure 2.
  8. Line 248, why ADAM optimizer? Needs reference of better performance or quantification of use.
  9. As training and ablation studies are performed with synthetically noised images, in which added noise shows a regular pattern, I am mostly interested in real oisy images, thus it would be better to have more visuals on that section (Section 4.3)

Reviewer 3 Report

This article introduces a dense residual transformer for solving low-level image denoising problem. At the heart of the model is a Sformer block, which is developed based on LeWin Transformer. The entire model shows promising performance, with even less number of parameters.

Overall, the article is well written, and the methodology is well motivated and reasonable. The results also demonstrate the effectiveness of the model. However, the article can be further improved by addressing some minor issues:

[1] It is not clear to me why LeWin is used to build the Sformer block. Why not using the vanilla attention module? More (theoretical or experimental) analysis should be given regarding this.

[2] The details of the preprocessing module in Eq.2 is not given. 

[3] Some important related work are missed. For example, Detail-Preserving Transformer for Light Field Image Super-Resolution, also propose a Transformer architecture for low-level vision problem. Please discuss it in the related work and clarify the difference. 

[4] For the ablation study, I will suggest to investigate the effect of the number of ETransformer block used in Sformer.

Round 2

Reviewer 3 Report

The revision has addressed all my concerns.